# Source-tracking ESBL-producing bacteria at the maternity ward of Mulago hospital, Uganda

Richard Mayanja[1,2‡], Adrian Muwonge[3☯]*, Dickson Aruhomukama[1,2], Fred Ashaba Katabazi[2], Mudarshiru Bbuye[2], Edgar Kigozi[2], Annettee Nakimuli[4‡], Musa Sekikubo[4‡], Christine Florence Najjuka[1☯]*, David Patrick Kateete[1,2☯]*

1 Department of Medical Microbiology, School of Biomedical Sciences, Makerere University College of Health Sciences, Kampala, Uganda, 2 Department of Immunology and Molecular Biology, School of Biomedical Sciences, Makerere University College of Health Sciences, Kampala, Uganda, 3 The Roslin Institute, College of Medicine and Veterinary Studies, The Royal (Dick) School of Veterinary Studies, University of Edinburgh, Easter Bush, Midlothian, United Kingdom, 4 Department of Obstetrics and Gynaecology, School of Medicine, Makerere University College of Health Sciences, Kampala, Uganda

☯ These authors contributed equally to this work.
‡ These authors also contributed equally to this work.
* adrian.muwonge@roslin.ed.ac.uk (AM); najjukafc@gmail.com (CFN); david.kateete@mak.ac.ug (DPK)

**Data Availability Statement:** All relevant data are within the paper and its Supporting Information files.

## Abstract

### Introduction

*Escherichia coli*, *Klebsiella pneumoniae* and *Enterobacter* (EKE) are the leading cause of mortality and morbidity in neonates in Africa. The management of EKE infections remains challenging given the global emergence of carbapenem resistance in Gram-negative bacteria. This study aimed to investigate the source of EKE organisms for neonates in the maternity environment of a national referral hospital in Uganda, by examining the phenotypic and molecular characteristics of isolates from mothers, neonates, and maternity ward.

### Methods

From August 2015 to August 2016, we conducted a cross-sectional study of pregnant women admitted for elective surgical delivery at Mulago hospital in Kampala, Uganda; we sampled (nose, armpit, groin) 137 pregnant women and their newborns (n = 137), as well as health workers (n = 67) and inanimate objects (n = 70 –beds, ventilator tubes, sinks, toilets, door-handles) in the maternity ward. Samples (swabs) were cultured for growth of EKE bacteria and isolates phenotypically/molecularly investigated for antibiotic sensitivity, as well as β-lactamase and carbapenemase activity. To infer relationships among the EKE isolates, spatial cluster analysis of phenotypic and genotypic susceptibility characteristics was done using the Ridom server.

### Results

Gram-negative bacteria were isolated from 21 mothers (15%), 15 neonates (11%), 2 health workers (3%), and 13 inanimate objects (19%); a total of 131 Gram-negative isolates were

**Funding:** The author(s) received no specific funding for this work.

**Competing interests:** The authors have declared that no competing interests exist.

identified of which 104 were EKE bacteria i.e., 23 (22%) *E. coli*, 50 (48%) *K. pneumoniae*, and 31 (30%) *Enterobacter*. Carbapenems were the most effective antibiotics as 89% (93/104) of the isolates were susceptible to meropenem; however, multidrug resistance was prevalent i.e., 61% (63/104). Furthermore, carbapenemase production and carbapenemase gene prevalence were low; 10% (10/104) and 6% (6/104), respectively. Extended spectrum β-lactamase (ESBL) production occurred in 37 (36%) isolates though 61 (59%) carried ESBL-encoding genes, mainly $bla_{CTX-M}$ (93%, 57/61) implying that $bla_{CTX-M}$ is the ideal gene for tracking ESBL-mediated resistance at Mulago. Additionally, spatial cluster analysis revealed isolates from mothers, new-borns, health workers, and environment with similar phenotypic/genotypic characteristics, suggesting transmission of multidrug-resistant EKE to new-borns.

## Conclusion

Our study shows evidence of transmission of drug resistant EKE bacteria in the maternity ward of Mulago hospital, and the dynamics in the ward are more likely to be responsible for transmission but not individual mother characteristics. The high prevalence of drug resistance genes highlights the need for more effective infection prevention/control measures and antimicrobial stewardship programs to reduce spread of drug-resistant bacteria in the hospital, and improve patient outcomes.

## Introduction

The World Health Organization (WHO) estimates that 5 million neonatal deaths occur annually, disproportionately affecting populations in the developing countries. Septicaemia is among the leading causes of morbidity and mortality in neonates and infants in the developing countries [1]. In addition to causing common skin and urinary tract infections, members of the *Enterobacteriaceae* family, especially *Escherichia coli*, *Klebsiella pneumoniae*, and *Enterobacter* species (spp.) (herein EKE bacteria) are reported to be the leading cause of septicaemia in Africa [2]. Clinicians increasingly recognise septicaemia as a life-threatening condition due to organ failure resulting from host deregulations and cellular metabolic breakdown [3]. Therefore, immediate clinical management is needed, which is dominated by the use of β-lactam class of antibiotics, especially the extended-spectrum β-lactam agents like the third-generation cephalosporins (e.g., ceftriaxone, ceftazidime). However, Gram-negative bacteria, especially members of the *Enterobacteriaceae* family, have increasingly become resistant to third-generation cephalosporins [4]. This has made infections they cause increasingly difficult to manage. Phenotypically, drug resistant *E. coli*, *K. pneumoniae*, and *Enterobacter* spp. produce β-lactamases that block the action of antibiotics; genotypically, these bacteria harbour extended spectrum β-lactamase (ESBL) encoding genetic elements like $bla_{CTX-M}$, $bla_{TEM}$, and $bla_{SHV}$, as well as AmpC encoding genes like *DHA*, *CMY*, and *CIT*. ESBL-mediated resistance is prevalent among the *Enterobacteriaceae* in African settings, for example in Uganda and Tanzania, where management using cephalosporins has been reported [5]. Moreover, ESBL-producing *Enterobacteriaceae* carry additional genetic elements like $bla_{VIM}$, $bla_{IMP}$, $bla_{KPC}$, $bla_{OXA-48}$, and $bla_{NDM}$, which encode carbapenemases i.e., VIM (veronica integrin Metallo-beta-lactamases), IMP (imipenemase), KPC (*Klebsiella pneumoniae* carbapenemase), OXA-48 (oxacillinase-48), and NDM-1 (New Delhi Metallo-beta-lactamase-1), respectively [6]. The

carbapenemases hydrolyse almost all β-lactam antibiotics [7] and enable resistance to carbapenems, a group of highly effective antibiotics [8].

Antibiotic resistant bacteria become ubiquitous when susceptible sub-populations that do not carry resistance genes are exposed to antibiotics that kill susceptible bacteria, thus selecting for resistant populations [9]. Furthermore, it is widely accepted that a history of visiting a hospital is a risk factor for acquisition of ESBL-producing Gram-negative bacteria, which are known for colonizing hospital surfaces, health workers, and pregnant women accessing prenatal hospital services [10,11]. This inherently makes neonates/infants an extremely high-risk group. Over the last 20 years, Mulago National Referral Hospital in Kampala, Uganda, has registered a considerable increase in neonatal morbidity and mortality predominantly caused by Gram-negative bacteria [12]. Moreover, studies conducted in Uganda have shown that ESBL-producing isolates are highly resistant to third-generation cephalosporins, specifically ceftazidime and cefotaxime [3], and that ESBL production occurs at variable levels at the hospital [5,12]. This, coupled with their ability to persist in hospital environments, makes ESBL-producing bacteria a significant health risk to neonates [13].

DNA amplification techniques combined with conventional phenotypic characterization of antibiotic resistance allows not only to cost-effectively ascribe phenotypic resistance to responsible genes, but also supports source-tracking at the hospital and community level in a resource-limited setting [14–16]. In this study, we examined the dispersal of ESBL-producing EKE bacteria in the maternity ward at Mulago hospital using phenotypic and genotypic characteristics of the isolates, and identified the potential source of drug resistant bacteria for neonates. This is critical for understanding the clinical and sanitary points of control, hence contributing to a reduction in hospital-acquired antimicrobial resistance.

## Methods

### Study setting

The study was conducted at Mulago hospital in Kawempe division, 3 km from Kampala city centre. Mulago serves as both the national referral hospital for Uganda and a teaching hospital for Makerere University; it is the largest public hospital in the country with 1,600 beds and a 1:40 doctor-to-patient ratio. It receives about 100 pregnant women daily, delivering up to 60 babies by ~50 midwives. Nearly half of these babies are born by Caesarean section [17]. According to hospital records there were 31,201 babies born in 2010, 33,331 in 2011, 33,231 in 2012 and 31,400 in 2013; in 2014, 30,000 babies were delivered at the hospital, which is about 68% of all the babies born in Uganda, giving it a claim to the top position of the busiest labour wards in the world [17].

### Study design, participants and eligibility criteria

The study design was cross-sectional, centred around the routine maternity activities at the hospital. The study analysed *Enterobacteriaceae* isolates cultured from samples (swabs) collected in a parallel study that looked at community methicillin resistant *Staphylococcus aureus* (MRSA) carriage and nosocomial MRSA acquisition among pregnant women in the maternity ward during August 2015 and August 2016, **Fig 1**. With consent, 137 pregnant women (purposive sampling) admitted to the hospital for elective surgical delivery (Caesarean section) were recruited, and subsequently were the babies delivered by the women. Samples (nasal, armpit and groin swabs) were collected from the pregnant women at admission, delivery and discharge from the hospital, **Fig 1**; additionally, samples were collected from neonates following surgical delivery. In case a mother or baby developed sepsis, swabs were collected from wounds or the vagina or baby's cord to investigate the cause of sepsis. Also, we collected 137

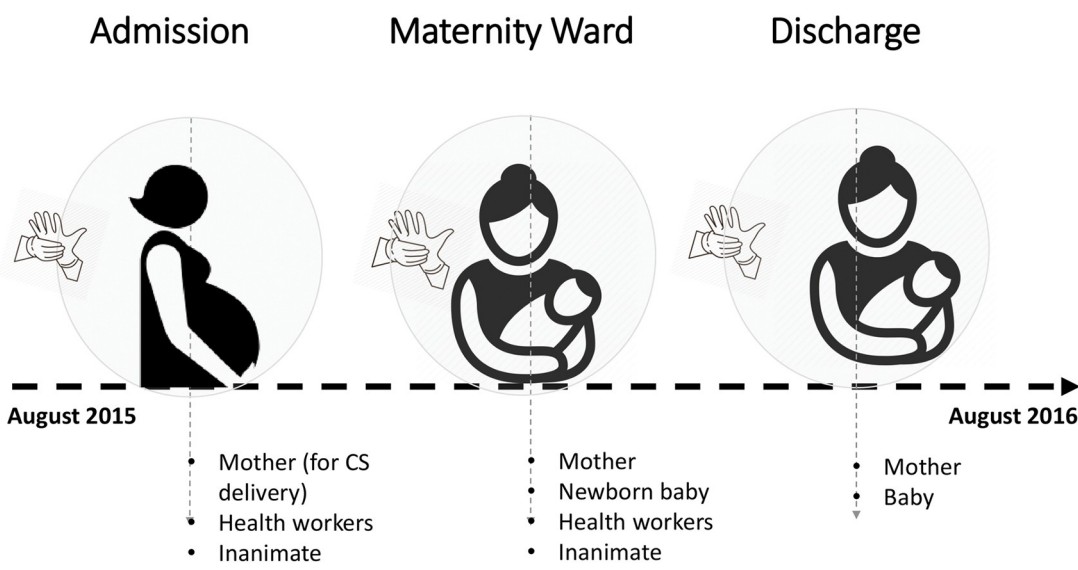

**Fig 1. Study schematic depicting participants and sampling timelines.**

environment samples–from health workers (n = 67, hereafter animate samples) who were handling the mothers/babies, as well as beds, ventilator tubes, sinks, toilets, and door-handles (n = 70, hereafter inanimate samples) in the labour ward. Overall, approximately 820 swab samples were processed and investigated for growth of *K. pneumoniae*, *E. coli* and *Enterobacter* spp.

The laboratory procedures were carried out in the Clinical Microbiology and Molecular Biology Laboratories of the College of Health Sciences, Makerere University. The Clinical Microbiology Laboratory participates in the College of American Pathologists' bacteriology external quality assurance scheme (CAP no. 7225593). In the laboratory, samples were inoculated on nonselective media (blood agar) and incubated overnight at 37˚C in ambient air; among the isolates obtained, a significant number (n = 167) with features suggestive of *Enterobacteriaceae* were identified and stored in 20% brain heart infusion (BHI)-glycerol at -20˚C. These are the isolates of interest that we retrieved and investigated; isolates were recovered by sub-culturing on blood agar at 37˚C in ambient air for 18–24 hours, and sub-culturing on MacConkey agar at 37˚C in ambient air for 12 hours. Identification to species level was based on phenotypic characteristics i.e., Gram staining and biochemical tests i.e., oxidase test, triple sugar iron agar (TSIA), indole, citrate utilization and urease production tests [18].

### Antibiotic sensitivity testing

Antibiotic sensitivity testing was done with the Kirby-Bauer disc diffusion test [6] using sensitivity discs–ceftriaxone (CRO, 30 μg), cefotaxime (CTX, 30 μg), cefepime (FEP, 30 μg), ceftazidime (CAZ, 30 μg), cefoxitin (FOX, 30 μg), cefoxitin/cloxacillin (FOX/CLOX, 30 μg / 200 μg/ml), meropenem (MEM, 10 μg), meropenem/ethylenediaminetetraacetic acid (MEM/EDTA, 30 μg/100 μg/ml), ciprofloxacin (CIP, 5 μg), gentamicin (CN, 10 μg), chloramphenicol (C, 30 μg), tetracycline (TE, 30 μg), trimethoprim/sulfamethoxazole (SXT, 1.25/23.75μg), amoxicillin-clavulanate (AMC, 30 μg), aztreonam (ATM, 30 μg), and piperacillin-tazobactum (TPZ, 110 μg). Briefly, an inoculum was prepared from a pure culture plate of a test isolate grown overnight. This was done by touching with a sterile loop the top of 3-to-5 colonies with similar appearance, suspending in normal saline and adjusting turbidity to 0.5 McFarland

(approximately $1.5 \times 10^8$ colony forming units [CFU]). Adjusting the density of the test suspension to that of the standard was done by adding more bacterial suspension or sterile normal saline. A sterile cotton swab was dipped into the bacterial suspension, and excess liquid removed by rotating the swab several times with firm pressure on the inner wall of the tube above the fluid level. Using the swab, a Mueller Hinton Agar (MHA) plate was streaked to form a bacterial lawn. To obtain uniform growth, the plate was streaked with the swab in one direction, rotated at 60 degrees and streaked again in another direction. The rotation was repeated three times then the swab passed round the edge of the agar surface as it was drawn across the plate. The plate was allowed to airdry for about 3–5 minutes before adding the antibiotic disc. Using a sterile pair of forceps, the antibiotic disc was added to the media plate and gently pressed on the agar to ensure it was attached. MHA plates with antibiotic discs were incubated at 37°C overnight in ambient air, after which zones of inhibition (in mm) were measured using a divider and ruler, and interpreted according to the Clinical and Laboratory Standards Institute (CLSI) guidelines (2015) [19].

### ESBL screening

Isolates with inhibition zone diameters suggestive of ESBL production i.e., ceftriaxone (CRO) = 23 mm, cefotaxime (CTX) = 26 mm, aztreonam (ATM) = 21 mm, and ceftazidime (CAZ) = 21 mm [20] were screened for ESBL production using the double disc synergy test and the modified double disc synergy test (MDDST), in which cefepime (FEP) replaced ceftriaxone [21]. An amoxicillin-clavulanate disc (20/10 μg) along with four cephalosporins discs i.e., cefotaxime, ceftriaxone, ceftazidime, and cefepime, were used. A lawn culture of the test isolate was made on an MHA plate with an amoxicillin-clavulanate disc placed in the centre of the plate. Then, cefotaxime, ceftriaxone, ceftazidime, and cefepime discs were placed 20 mm centre-to-centre to the amoxicillin-clavulanate disc and incubated overnight at 37°C. Any distortion or increase in the zone of clearance towards the amoxicillin-clavulanate disc was considered positive for ESBL production; *K. pneumoniae* strain 700603 and *E. coli* strain 25922 were used as the positive and negative controls, respectively.

### Screening for AmpC enzymes

Isolates with a cefoxitin inhibition zone diameter of ≤17 mm were screened for AmpC enzyme production using cefoxitin disc (30 μg) and cefoxitin (30 μg) + cloxacillin (200 μg) discs on MHA plates incubated overnight at 37°C. The inhibition zone diameter around the cefoxitin + cloxacillin disc was compared to that of cefoxitin without cloxacillin for confirmation of AmpC β-lactamase production. An inhibition zone diameter difference of ≥4 mm was interpreted as positive for AmpC production. Cloxacillin was used as the inhibitor for AmpC enzyme activity, while *E. coli* strain ATCC25922 was used as the negative control [6].

### Screening for carbapenemases

Isolates with a meropenem (10 μg) inhibition zone diameter of ≤23mm were screened for carbapenemase production using the modified Hodge's test (MHT) [22]. A 1:10 dilution of the indicator/susceptible organism (*E. coli* ATCC 25922) was adjusted to turbidity equivalent to 0.5 McFarland in normal saline, streaked on MHA plate and air-dried for 5–10 minutes, and a meropenem disc (10 μg) placed in the centre of the plate. Test isolates were streaked outward from the disc to the edge of the plate (20–25 mm in length) using a sterile swab. The same procedure was carried out for the positive control (*K. pneumoniae* ATCC® BAA-1705™) and the negative control (*K. pneumoniae* ATCC® BAA-1706™). Plates were incubated in ambient air for 16–20 hours at 37°C and results interpreted according to the CLSI (2015) guidelines.

Briefly, β-lactamase production was verified based on distortion of the inhibition zone; a positive result had enhanced growth around the positive control streak at the intersection of the zone of inhibition (i.e., formation of a clover-leaf indentation of indicator strain growing along the streak of the test organism within the antimicrobial diffusion diameter); on the other hand, a negative result had no growth of the indicator strain along the streak of the test isolate within the disc diffusion zone.

To screen for metallo-beta-lactamase production, an overnight culture of a test isolate equivalent to 0.5 McFarland was inoculated on MHA plates using a sterile swab. After 5–10 minutes of drying, two meropenem discs (10 μg) were placed on the surface of the agar 15 mm apart, centre-to-centre. Ten microliters of 0.5 M EDTA was added to one of the meropenem discs and incubated at 37°C overnight. An increase in the zone of inhibition by ≥5 mm around the EDTA potentiated disc was interpreted as positive for metallo-β-lactamase production.

### Detection of antibiotic resistance genetic elements

Isolates screened for ESBL and carbapenemase activity were molecularly investigated for ESBL- and carbapenemase gene carriage. We used conventional PCR to detect $bla_{CTX-M}$, $bla_{TEM}$, and $bla_{SHV}$ genes which confer bacterial resistance to β-lactam agents except carbapenems and cephamycin [8]. We also used PCR to detect carbapenemase encoding genes $bla_{VIM}$, $bla$IMP, and $bla_{NDM}$. Except for the $bla_{CTX-M}$ gene variants where we used inhouse primers, we used previously published primers and conditions for the PCRs [6,23], **S1 Table**. PCR amplicons were analysed by gel electrophoresis on a 1.5% agarose gel stained with ethidium bromide and viewing DNA bands in a UV trans-illuminator. Isolates that were previously confirmed to be positive or negative for the genes being investigated were used as positive and negative controls, respectively. Furthermore, PCR-amplicons were sequenced to confirm the resistance genes through BLAST searches at the National Centre for Biotechnology Information (NCBI) https://blast.ncbi.nlm.nih.gov/Blast.cgi

### Data analysis

Disc diffusion using the Kirby Bauer method was interpreted according to CLSI guidelines [19]. Microsoft Excel 2016 and SPSS version 16.0 were used for data entry and statistical analyses. Differences in proportions and means were compared using chi square and the student t-test, respectively. A p-value of <0.05 was considered statistically significant. Cluster analysis of the phenotypic and genotypic characteristics of the isolates was done using Ridom GmBH, Münster, Germany. Here, the phylogenetic analysis module was used to cluster phenotypes or genotypes by similarity of profiles and then visualized using UPGMA phylogenetic tree.

### Ethical considerations

Ethical approval was provided by the School of Biomedical Sciences Research and Ethics Committee at Makerere University (SBS-REC 434); a waiver of consent to use archived samples was provided by the SBS-REC. Authors did not have access to information that could identify individual participants during or after data collection.

## Results and discussion

### Participants' demographics and bacterial isolates

In this study, purposive sampling was used to enrol 137 pregnant women who underwent Caesarean surgical delivery (C-section) and their natal babies (n = 137 –there were no multiple

**Table 1. Demographics of pregnant women (n = 137) who underwent Caesarean surgical delivery and screened for isolation of *E. coli*, *K. pneumoniae* and *Enterobacter* spp.**

| Variable | Categorization | n (%) |
|---|---|---|
| Highest education level attained | Primary | 30 (22) |
| | Secondary or Vocational | 65 (47) |
| | Advanced level | 25 (18) |
| | Tertiary institute* | 9 (7) |
| | Bachelor's degree | 8 (6) |
| Sample site (and time point) | Groin (on admission) | 23 (17) |
| | Groin (on discharge) | 58 (42) |
| | Armpit (on admission) | 0 |
| | Armpit (on discharge) | 32 (23) |
| | Nose (on admission) | 24 (18) |

* Tertiary but not university.

pregnancies), and screened for contamination with *E. coli*, *K. pneumoniae* and *Enterobacter* spp. Key demographics of the mothers are summarized in **Table 1**. Overall, Gram-negative bacteria were isolated from 21 mothers, 15 babies (neonates), 2 health workers, and 13 inanimate objects, **Table 2**. A total of 131 Gram-negative isolates were identified, of which 104 isolates were of our interest i.e., *E. coli*, *K. pneumoniae* and *Enterobacter* spp.; *K. pneumoniae* was the most prevalent species (38%, 50/131) followed by *Enterobacter* spp. (24%, 31/131), **Table 2**. Other Gram-negative bacteria were identified but not discussed further–these include *Citrobacter* spp., *Pseudomonas* spp., and *Acinetobacter* spp., **Table 3**.

## Antibiotic susceptibility profiles

The highest drug sensitivity level was noted for carbapenem antibiotics whereby 97% (30/31) of *Enterobacter* spp., 89% (44/50) of *K. pneumoniae*, and 82% (19/23) of *E. coli* isolates were susceptible to meropenem, **Table 4**. Nevertheless, 58% (29/50) of the *K. pneumoniae* isolates, 70% (16/23) of *E. coli*, and 58% (18/31) of *Enterobacter* spp. were multidrug resistant; to determine multidrug resistance (MDR), resistance to β-lactams, aminoglycosides, trimethoprim-sulfamethoxazole, tetracyclines and fluoroquinolones was considered. **Table 5** depicts the resistance combinations noted and virtually all patterns involved a non-β-lactam agent; the most common MDR pattern among the *K. pneumoniae* isolates was gentamicin+-chloramphenicol+trimethoprim-sulfamethoxazole while for *E. coli* and *Enterobacter* it was ciprofloxacin+gentamicin+chloramphenicol+tetracycline+trimethoprim-sulfamethoxazole.

**Table 2. *E. coli*, *K. pneumoniae* and *Enterobacter* spp. isolated and investigated (n = 104).**

| Source | *E. coli* (%) | *K. pneumoniae* (%) | *Enterobacter spp.* (%) | Total* |
|---|---|---|---|---|
| Mother, 21/137 (15%) | 05 (4) | 18 (13) | 07 (5) | **30** |
| Baby, 15/137 (11%) | 05 (4) | 15 (11) | 10 (7) | **30** |
| Environment (animate): 2/67 (3%) | 10 (15) | 06 (9) | 02 (3) | **18** |
| Environment (inanimate): 13/70 (19%) | 03 (1) | 11 (16) | 12 (17) | **26** |
| Total (%) | **23 (22)** | **50 (48)** | **31 (30)** | **104** |

*Some participants/objects grew multiple bacterial species (i.e., polymicrobial samples).

**Table 3. Other Gram-negative bacteria identified, n = 131 (%).**

| Source | Citrobacter | Acinetobacter | Pseudomonas | K. oxytoca | Total |
|---|---|---|---|---|---|
| Mothers (n = 137) | 7 (5) | 5 (4) | 2 (2) | 0 | **14** |
| Babies (n = 137) | 2 (2) | 6 (5) | 1 (0.8) | 1 (0.8) | **10** |
| Health workers (n = 67) | 6 (5) | 7 (5) | 2 (2) | 0 | **15** |
| Environment (Inanimate) (n = 70) | 2 (2) | 0 | 0 | 0 | **2** |
| **Total** | **17 (13)** | **18 (14)** | **5 (4)** | **1 (0.8)** | **41** |

## Characterisation of beta-lactamases and carbapenemases

ESBL, AmpC and carbapenemase activity was detected in all the three species, **Table 6**; ESBL activity was highest in *K. pneumoniae* (50%) while AmpC activity was highest in *Enterobacter* spp. (45%). However, carbapenemase activity was comparatively low, **Table 6**.

## Characteristics of genetic determinants of antibiotic resistance

**Tables 7** and **S2** depict the frequency and distribution of the antibiotic resistance genetic elements associated with resistance to β-lactams and carbapenems. Overall, ESBL and carbapenemase encoding genes were detected in 59% (61/104) of the isolates and the former were more prevalent particularly the $bla_{CTXM-U/15}$ gene. While $bla_{CTX-M}$, $bla_{TEM}$ and $bla_{SHV}$ occurred in isolates regardless of the presence of phenotypic ESBL-activity, $bla_{VIM}$, $bla_{NDM}$ and $bla_{IMP}$ occurred in only carbapenemase-producing isolates. Furthermore, $bla_{CTX-M-15}$ was the only $bla_{CTX-M}$ gene found in *E. coli* and *Enterobacter* spp. while it occurred in 23 of the 29 ESBL gene positive *K. pneumoniae* isolates, implying that the six *K. pneumoniae* isolates with the universal $bla_{CTX-M-U}$ gene carried other $bla_{CTX-M}$ types. Overall, these data show that $bla_{CTX-M-15}$ is a predominant ESBL encoding gene in this setting. Furthermore, carriage of multiple resistance genetic elements was frequent in *K. pneumoniae* and *Enterobacter* spp., especially ESBL encoding genes and the most common pattern was $bla_{CTXM-U/15} + bla_{SHV}$, implying that the $bla_{TEM}$ and $bla_{SHV}$ genes in this setting are co-transmitted with $bla_{CTXM-U/15}$ in that carriage of $bla_{TEM}$ alone or $bla_{SHV}$ alone wasn't seen. On the other hand, the carbapenemase encoding genes were less prevalent and occurred in only six isolates, **S2 Table**. Note, while the carbapenemase gene prevalence is low in this study, four of the six carbapenemase gene positive isolates (i.e., $bla_{VIM}+$, $bla_{IMP}+$, $bla_{NDM}+$) co-carried the genes, and almost all were *ESBL* gene positive, **S2 Table**.

**Table 4. Antibiotic susceptibility characteristics of the EKE bacterial isolates n, (%)*.**

| Species | AMC | TPZ | CRO | CTX | CAZ | ATM | FEP | FOX | MEM | SXT | CIP | CN | C | TE |
|---|---|---|---|---|---|---|---|---|---|---|---|---|---|---|
| *E. coli* (n = 23) | 16 (70) | 14 (61) | 8 (35) | 8 (35) | 4 (17) | 10 (44) | 4 (17) | 12 (52) | 19 (83) | 10 (44) | 5 (22) | 10 (44) | 14 (61) | 9 (39) |
| *K. pneumoniae* (n = 50) | 30 (60) | 25 (50) | 18 (36) | 13 (26) | 21 (38) | 21 (42) | 18 (36) | 41 (82) | 44 (88) | 13 (26) | 5 (10) | 26 (52) | 19 (38) | 36 (72) |
| Enterobacter (n = 31) | 10 (32) | 18 (58) | 8 (26) | 13 (42) | 10 (32) | 13 (42) | 14 (45) | 2 (7) | 30 (97) | 14 (45) | 20 (65) | 11 (36) | 15 (48) | 20 (65) |

AMC, ampicillin-sulbactam; TPZ, piperacillin-tazobactam; CRO, ceftriaxone; CTX, cefotaxime; CAZ, ceftazidime; ATM, aztreonam; FEP, meropenem; FOX, cefoxitin; MEM, meropenem; SXT, trimethoprim-sulfamethoxazole; CIP, ciprofloxacin; CN, gentamicin; C, chloramphenicol; TE, tetracycline.
*Refers to percentage of drug susceptible isolates.

**Table 5. Multiple resistance patterns to classes of antibiotics.**

| Combination | *K. pneumoniae* n = 50 (%) | *E. coli* n = 23 (%) | Enterobacter spp., n = 31 (%) |
|---|---|---|---|
| C+SXT | 5 (10%) | 1 (4%) | 0 |
| C+TE | 1 (2%) | 0 | 0 |
| C+CN | 0 | 0 | 2 (6%) |
| CN+SXT | 0 | 0 | 1 (3%) |
| CIP+SXT | 0 | 1 (4%) | 0 |
| CIP+TE | 2 (4%) | 0 | 0 |
| TE+SXT | 1 (2%) | 0 | 1 (3%) |
| CN+C+SXT | 9 (18%) | 0 | 0 |
| CN+TE+SXT | 0 | 1 (4%) | 1 (3%) |
| CN+C+SXT | 0 | 0 | 3 (10%) |
| CIP+C+SXT | 2 (4%) | 0 | 0 |
| CIP+TE+SXT | 1 (2%) | 4 (17%) | 0 |
| CIP+CN+SXT | 0 | 1 (4%) | 0 |
| C+TE+SXT | 1 (2%) | 1 (4%) | 0 |
| CIP+CN+TE+SXT | 0 | 0 | 1 (3%) |
| CIP+CN+C+SXT | 0 | 0 | 1 (3%) |
| CIP+C+TE+SXT | 0 | 1 (4%) | 0 |
| CN+C+TE+SXT | 5 (10%) | 1 (4%) | 2 (6%) |
| CIP+CN+C+TE+SXT | 2 (4%) | 5 (22%) | 6 (19%) |

SXT, trimethoprim-sulfamethoxazole; CIP, ciprofloxacin; CN, gentamicin; C, chloramphenicol; TE, tetracycline.

## Inferring transmission from clustering of drug resistance phenotypes and genotypes

For an insight into the source/transmission of MDR *E. coli*, *K. pneumoniae* and *Enterobacter* spp. in the maternity ward, spatial cluster analysis of phenotypic and genotypic susceptibility characteristics was performed and the analyses presented as dendrograms for inferring relationships, **Figs 2** & **3**. Based on phenotypic susceptibility characteristics, seven clusters comprising two or more isolates from mothers, their babies, health workers (animate), and/or environment (inanimate) were noted, **Fig 2**. Additionally, isolates from mothers with susceptibility characteristics similar to isolates from babies that were not their own were noted. Still, based on molecular susceptibility characteristics, eight clusters comprising two to eight isolates from mothers, their babies, health workers (animate), and/or environment (inanimate) were noted, **Fig 3**. Overall, these data allude to occurrence of epidemiological links for the clustered isolates hence, transmission in the maternity ward of MDR *E. coli*, *K. pneumoniae* and *Enterobacter* spp. from mothers to new-borns.

Overall, this study depicts a high recovery of *K. pneumoniae*, *Enterobacter* and *E. coli* with phenotypic and genotypic characteristics of multi-resistance in the maternity ward of Mulago

**Table 6. Prevalence of beta-lactamases and carbapenemases.**

| Species | ESBLs alone (%) | AmpC alone (%) | ESBL+AmpC (%) | Carbapenemases (%) |
|---|---|---|---|---|
| *E. coli* (n = 23) | 9 (39) | 5 (22) | 6 (26) | 3 (13) |
| *K. pneumoniae* (n = 50) | 25 (50) | 6 (12) | 3 (6) | 6 (12) |
| Enterobacter spp. (n = 31) | 3 (10) | 14 (45) | 12 (39) | 1 (3) |

**Table 7. Summary of the antibiotic resistance genes among PCR-positive isolates.**

| Species | $bla_{CTXM-U}$ | $bla_{CTXM-15}$ | $bla_{TEM}$ | $bla_{SHV}$ | $bla_{VIM}$ | $bla_{IMP}$ | $bla_{NDM}$ |
|---|---|---|---|---|---|---|---|
| E. coli n = 23 (%) | 12 (52) | 12 (52) | 7 (30) | 3 (13) | 02 (9) | 02 (9) | 02 (9) |
| Klebsiella n = 50 (%) | 29 (58) | 23 (46) | 10 (20) | 23 (46) | 02 (4) | 01 (2) | 0 |
| Enterobacter n = 31 (%) | 16 (52) | 16 (52) | 09 (29) | 3 (10) | 01 (3) | 0 | 0 |
| **Total** | **57** | **51** | **26** | **29** | **05** | **03** | **02** |

hospital. The clustering of phenotypic and genotypic profiles by time and space suggests active transmission between mother and new-born babies, as well as health workers and their maternity ward environment. This calls into question the effectiveness of infection prevention and control strategies, given the isolation of these potential pathogens from healthcare equipment, ward environment and the patients. However, the fact that bacteria were also isolated from participants on admission into the hospital, there is an indication that some of the profiles could be acquired from the community before the mothers are admitted, suggesting a role of community as a contributor to the diversity of organisms observed in this study, **Fig 4**.

In addition, the high prevalence of *K. pneumoniae*, *Enterobacter* and *E. coli* in mothers and their natal babies depict the hygiene levels of items mothers use on hospital admission. Our findings are in line with studies in similar settings for example, Kayange *et al* (2010) who looked at the predictors of positive blood culture and deaths among neonates with suspected neonatal sepsis in a tertiary hospital in Mwanza, Tanzania [1]. Also, in Kenya a study that investigated hospital acquired infections in a private pediatrics' hospital found *K. pneumoniae* to be the most prevalent species followed by *Pseudomonas aeruginosa* and *Enterobacter cloacae* [24].

Moreover, the high MDR levels among isolates in this study suggests high selection pressure in the hospital [25], as well as overuse and inappropriate use of antibiotics [26]. Indeed, there was high carriage of antibiotic resistance encoding genes (especially ESBL genes) among isolates, which means organisms have acquired resistance genes and disseminated them to other organisms, for example, through plasmids that can carry various genes [27]. Among the ESBL-encoding genes, $bla_{CTX-M}$, $bla_{SHV}$ and $bla_{TEM}$ were detected at rates comparable to previous studies [28]. Though, we found $bla_{CTX-M}$ to be the most frequent gene while previous studies in similar settings reported $bla_{TEM}$ (48.7%) to be the most prevalent, followed by $bla_{CTX-M}$ (7.6%) and $bla_{SHV}$ (5.1%) [28]. We also found carriage of antibiotic resistant isolates by health workers to be comparable to that of studies done elsewhere [29–31]. Health workers can acquire bacterial contamination by direct contact with patients, body fluids secretions, or touching contaminated environmental surfaces within the hospital environment [32]. Just like in earlier studies in Uganda [6,23,33], the carbapenemase gene prevalence remains low at Mulago hospital; despite this, we identified carriage of more than one carbamenemase gene, suggesting enhanced drug resistance to carbapenems. Finally, spatial cluster analysis suggests transmission between animate and inanimate or a shared source of contamination for the maternity ward; this source could be items like health worker's gloves, stethoscopes and other items which were not sampled in our study [34].

## Limitations

The transmission investigation could have benefited from a more granular analytical method such as next generation sequencing to better track the source using single nucleotide polymorphisms. Due to resources this was not possible, nonetheless the findings provide clues that will

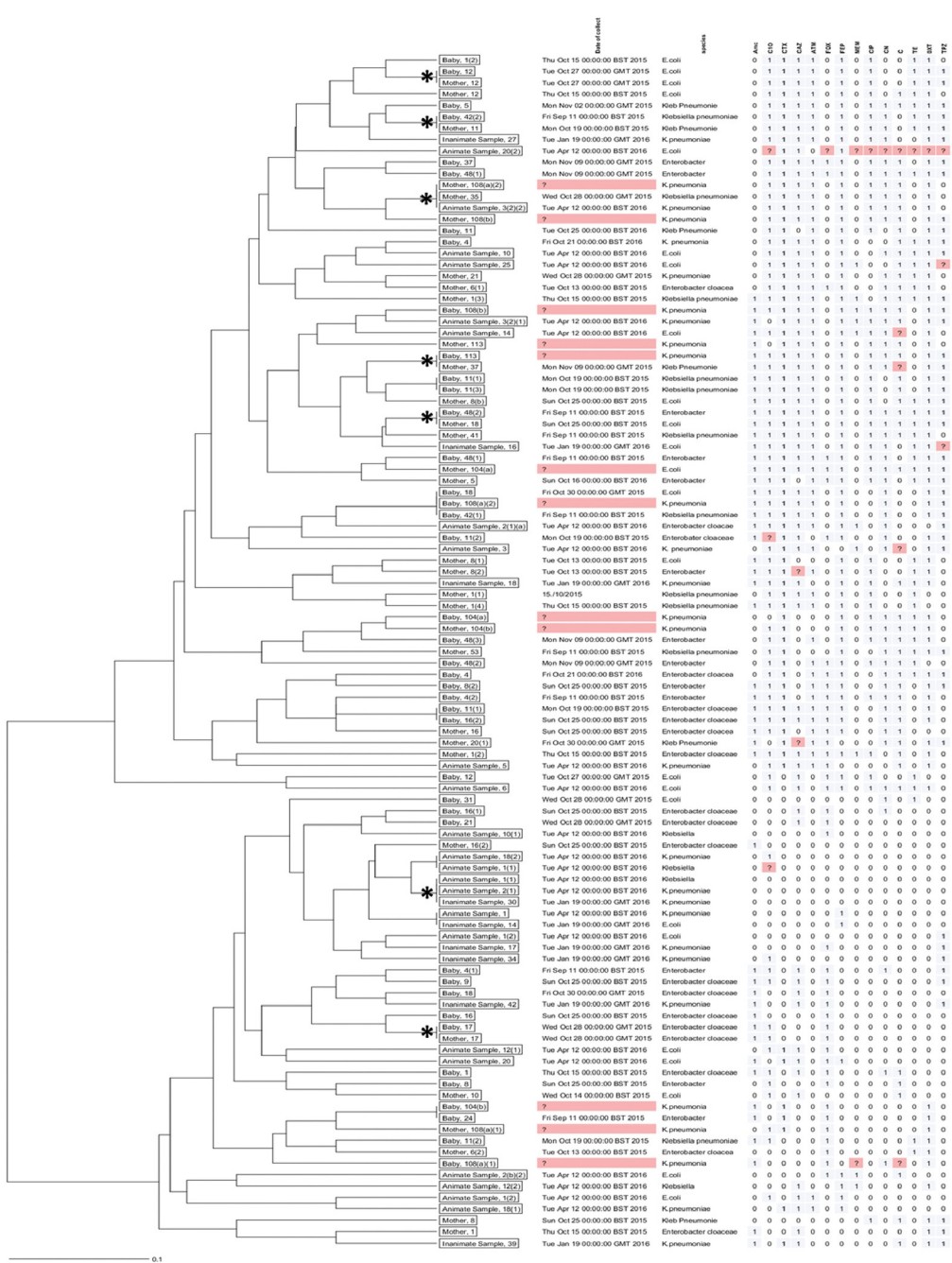

**Fig 2. Cluster analysis of phenotypic susceptibility characteristics.** Depicts seven clusters comprising drug resistant isolates of *E. coli*, *K. pneumoniae* and *Enterobacter* spp. with similar antibiotic susceptibility profiles hence, potential transmission of MDR bacteria from mothers/environment to new-borns. Clusters of isolates with similar susceptibility characteristics are denoted with an asterisk (*).

be further examined as and when resources become available. Also, the data utilized in the study was gathered in the period between 2015 and 2016, and considering that a lot of transformations might have taken place since then, it may not accurately depict the present circumstances at the hospital. In light of this, we suggest carrying out a similar study that employs updated datasets for a contemporary overview of the drug resistance situation at Mulago.

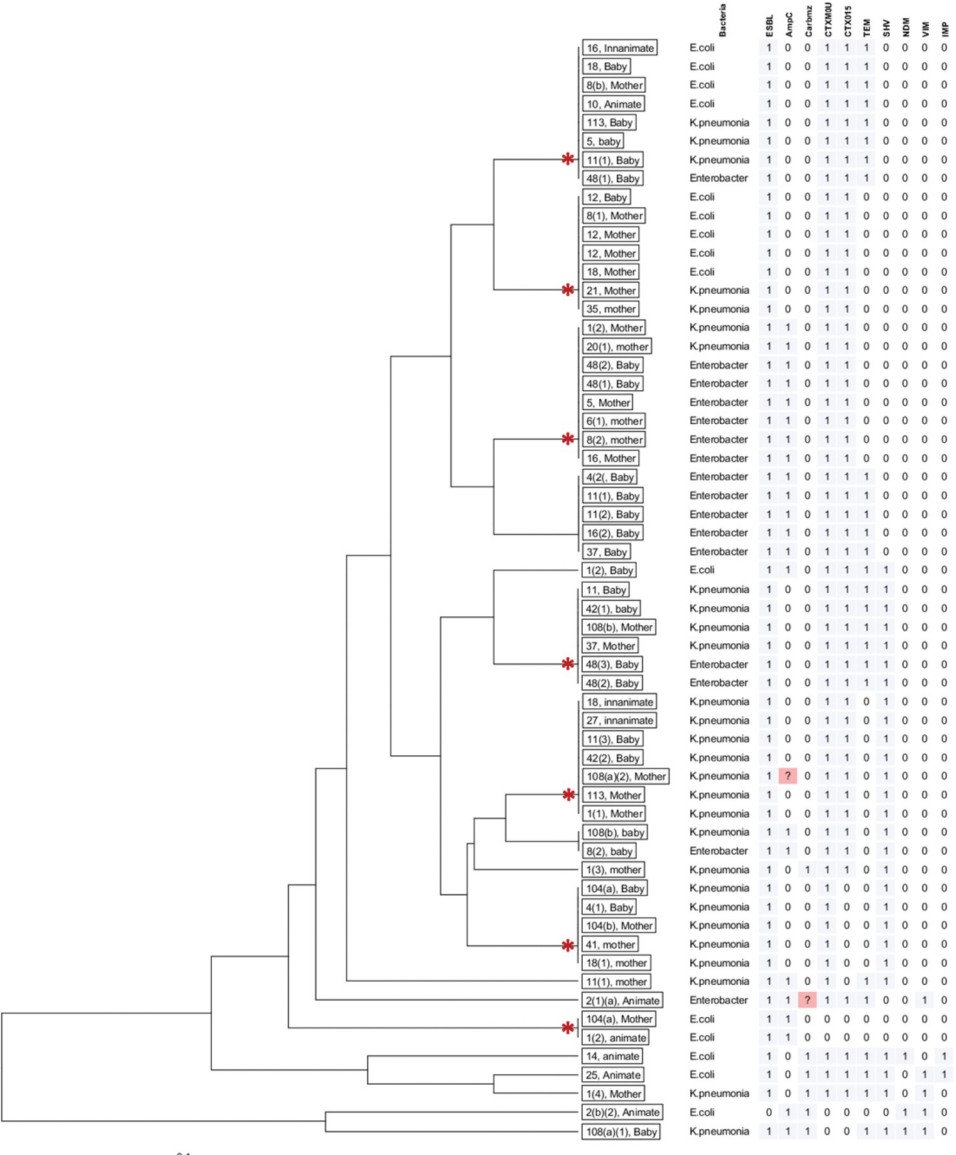

**Fig 3. Cluster analysis of genotypic susceptibility characteristics.** Depicts isolates of *E. coli*, *K. pneumoniae* and *Enterobacter* spp. with similar molecular susceptibility profiles hence, potential transmission of MDR bacteria from mothers to new-borns in the maternity ward. Asterisks (*) denote isolates from mothers, babies and/or environment with similar genotypic characteristics.

## Conclusions

Our findings suggest a potentially high exposure rate of mothers and their new-born babies to a variety of MDR *Enterobacteriaceae* strains when admitted for elective surgical delivery. The potential sources of these strains were health workers, maternity ward environment as well as introductions from the community by mothers. The findings suggest inadequacies in infection control practices on the maternity ward. Given the high prevalence of ESBLs on the ward, we recommend regular review of the infection control protocols in the maternity ward, and more studies should be conducted to look at other organisms and sample more items used in the

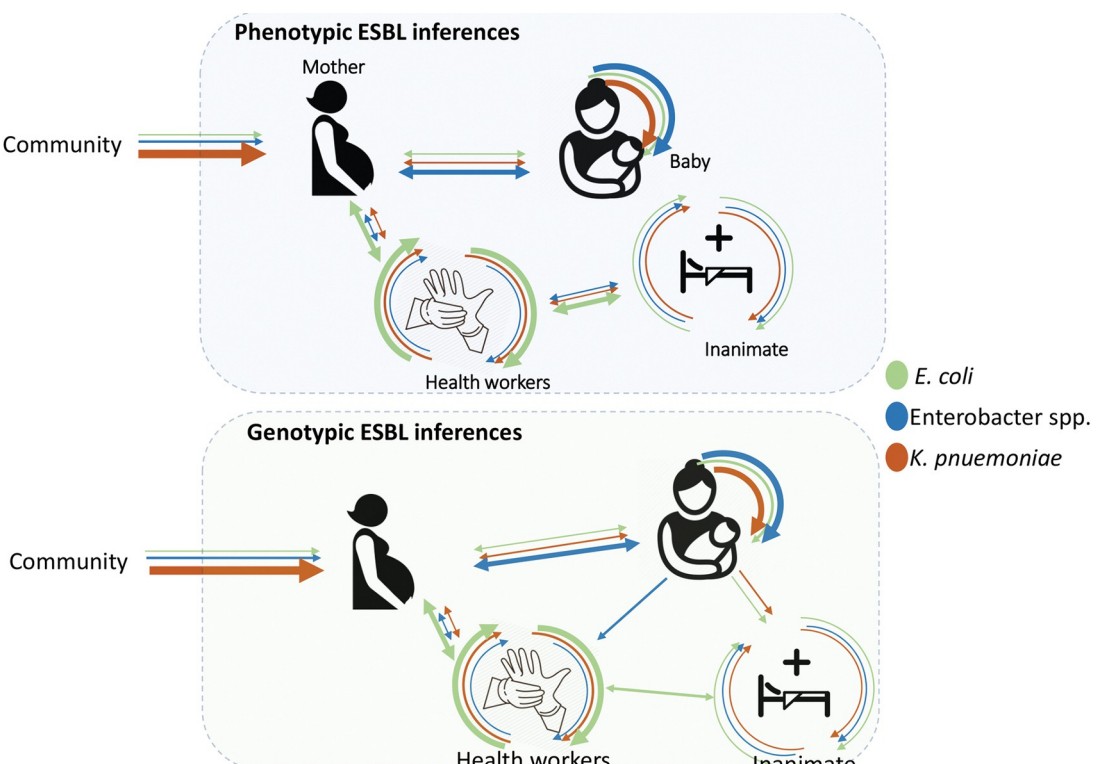

**Fig 4. Hypothetical sources of MDR *K. pneumoniae*, *E. coli* and *Enterobacter* spp. in the maternity ward of Mulago hospital.**

hospital especially those which are shared by patients. Molecular techniques with a high discriminatory power such as DNA sequencing and/or Pulsed Field Gel electrophoresis should be considered in future studies.

## Supporting information

**S1 Checklist. STROBE statement—Checklist of items that should be included in reports of observational studies.**
(DOCX)

**S1 Table. Primer sequences used to PCR-amplify ESBL-encoding and carbapenemase-encoding genes.**
(DOCX)

**S2 Table. Frequency and distribution of antibiotic resistance genetic elements among PCR-positive isolates.**
(DOCX)

## Acknowledgments

Special thanks to staff at the Departments of Immunology and Molecular Biology and Medical Microbiology, Makerere University College of Health Sciences for the support they provided during the time the research was conducted, as well as the research participants who agreed to be part of this study. Finally, special thanks to the Cambridge group that collected the isolates and made them available to us.

## Author Contributions

**Conceptualization:** Adrian Muwonge, Musa Sekikubo, Christine Florence Najjuka, David Patrick Kateete.

**Data curation:** Richard Mayanja, Adrian Muwonge.

**Formal analysis:** Adrian Muwonge, David Patrick Kateete.

**Investigation:** Annettee Nakimuli, Musa Sekikubo.

**Methodology:** Richard Mayanja, Dickson Aruhomukama, Fred Ashaba Katabazi, Mudarshiru Bbuye, Edgar Kigozi, Annettee Nakimuli, Musa Sekikubo, Christine Florence Najjuka.

**Resources:** Annettee Nakimuli.

**Software:** Adrian Muwonge.

**Supervision:** David Patrick Kateete.

**Validation:** Fred Ashaba Katabazi.

**Writing – original draft:** Richard Mayanja, Adrian Muwonge, David Patrick Kateete.

**Writing – review & editing:** Adrian Muwonge, Annettee Nakimuli, Musa Sekikubo, Christine Florence Najjuka, David Patrick Kateete.

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
