## [Decision Letter · Decision Letter 0]

3 Apr 2023

PONE-D-23-06934Source-tracking ESBL-producing bacteria at the maternity ward of Mulago Hospital, UgandaPLOS ONE

Dear Dr. Kateete,

Thank you for submitting your manuscript to PLOS ONE. After careful consideration, we feel that it has merit but does not fully meet PLOS ONE’s publication criteria as it currently stands. Therefore, we invite you to submit a revised version of the manuscript that addresses the points raised during the review process. Please submit your revised manuscript by May 18 2023 11:59PM. If you will need more time than this to complete your revisions, please reply to this message or contact the journal office at plosone@plos.org. Please include the following items when submitting your revised manuscript:A rebuttal letter that responds to each point raised by the academic editor and reviewer(s). You should upload this letter as a separate file labeled 'Response to Reviewers'.A marked-up copy of your manuscript that highlights changes made to the original version. You should upload this as a separate file labeled 'Revised Manuscript with Track Changes'.An unmarked version of your revised paper without tracked changes. You should upload this as a separate file labeled 'Manuscript'.If applicable, we recommend that you deposit your laboratory protocols in protocols.io to enhance the reproducibility of your results. Protocols.io assigns your protocol its own identifier (DOI) so that it can be cited independently in the future. For instructions see: https://journals.plos.org/plosone/s/submission-guidelines#loc-laboratory-protocols. Additionally, PLOS ONE offers an option for publishing peer-reviewed Lab Protocol articles, which describe protocols hosted on protocols.io. Read more information on sharing protocols at https://plos.org/protocols?utm_medium=editorial-email&utm_source=authorletters&utm_campaign=protocols.

We look forward to receiving your revised manuscript.

Kind regards,

Mabel Kamweli Aworh, DVM, MPH, PhD. FCVSN

Academic Editor

PLOS ONE

Journal Requirements:

   "Special thanks to staff at the Departments of Immunology and Molecular Biology and MedicalMicrobiology, Makerere University College of Health Sciences for the support they provided during the time the research was conducted, as well as the research participants who agreed to be part of this study. Finally, special thanks to the Cambridge group that collected the isolates and made them available to us. AM is a Chancellor’s Fellow at the Roslin Institute and his time was paid core funding and as a Future leader fellow funded by BBSRC (BB/P007767/1) and Wellcome Trust ISSF3 (IS3-R1.09 19/20)."

Reviewers' comments:

Reviewer's Responses to Questions

**Comments to the Author**

1. Is the manuscript technically sound, and do the data support the conclusions?

Reviewer #1: Yes

Reviewer #2: Yes

Reviewer #3: Yes

2. Has the statistical analysis been performed appropriately and rigorously? 

Reviewer #1: Yes

Reviewer #2: Yes

Reviewer #3: Yes

3. Have the authors made all data underlying the findings in their manuscript fully available?

Reviewer #1: Yes

Reviewer #2: Yes

Reviewer #3: Yes

4. Is the manuscript presented in an intelligible fashion and written in standard English?

Reviewer #1: Yes

Reviewer #2: Yes

Reviewer #3: Yes

5. Review Comments to the Author

Reviewer #1: This is a very informative and technical work. Few inputs made; work is good for publishing.

I appreciate the fact that you numbered your lines making reviewing this work a lot easier, though some numbered lines are blank I would use your numbering as is. Thank you

1. Line 34, page 9; please correct cross-section study to cross-sectional study

2. Lines 50 - 52, page 10; please review this statement and clarify

3. Lines 114 – 115, page 12; do you have access to more recent birth records? If yes, please use that unless you have a good reason for using these from 9 years ago.

4. Line 132, page 13; please do not abbreviate the word approximately

5. Line 137, page 14; please change carried out from to “carried out in “

6. Line 240, page 18; please use chi 2 or chi square

7. Lines 242 - 243, page 18; please change was to were

8. Include your keywords to the written manuscript

9. The omission of gloves used by health workers as part of where samples should be taken was a huge oversight in this work nevertheless much work was put into this study and it is very technical and highly informative. The work itself has a lot of papers embedded

10. Justice was done to the title

Reviewer #2: Thank you for this important study that characterizes the sources of ESBL which is crucial for infection control and prevention at a tertiary hospital ward in Uganda .

The Abstract is well written, however, the authors should consider adding the policy or practice implication to the abstract. Additionally, line 24 reads cross-section instead of cross-sectional.

Major strengths of this study include the well-written methodology and the presentation of the result findings in tables that are clear and easy to understand. A major weakness of this study is the time that the data was collected which was more than five years ago. however, the study still provides important insights.

The Introduction gives a good background and a good justification for the study.

Methods:

Line 108: Consider removing "isolates" as there is no description of isolates in this section.

Line 125: The authors should consider consistency in describing the sample collection sites e.g. nasal is referred to nares and anterior nares in different sections; armpit is referred to as axillar (line 264; also note that the correct word is axilla)

Line 127: The authors should consider changing "as well" to other phrases such as Also, Additionally etc.

Line 132: approx. should be be written in full or changed to another term. Same as for Line 161.

Line 173: Consider adding a 'comma' after "ruler"

Results and Discussion:

Line 251: The authors should consider writing "discussion" as a sentence case

Line 261: further should be in the past tense.

Line 328: The authors should consider changing "as well" to other phrases such as Also, Additionally etc.

Line 360: Kayange et al should be written as Kayange et al., and the year of the article should be included.

Limitations:

Line 387: The authors should consider writing the none the less as one word.

Also consider acknowledging that the data was collected in 2015-2016 and any implication if any.

Conclusions: Line 396: Consider reworking "carrying out more studies" to "more studies should be conducted". Also consider rephrasing "urgent infection control protocol" as this data for this study was collected between 2015-2016, and a lot of things might have changed.

Overall, this was a well conducted and reported study with necessary ethical approvals obtained.

Reviewer #3: The manuscript was well laid out with a good method section; However, authors need to explain how they arrived at a sample of 137 women? is it a purposive sampling technique? Religion of participants were listed in Table 1, and it is unclear how that relates to having ESBL producing bacteria. Consider dropping religion. Other minor edits as commented in the attached file

6. PLOS authors have the option to publish the peer review history of their article (what does this mean?). If published, this will include your full peer review and any attached files.

Reviewer #1: **Yes: **Folashade Onatola Bamidele

Reviewer #2: No

Reviewer #3: **Yes: **Augustine Olajide Dada

---

## [Author Response · Author response to Decision Letter 0]

9 May 2023

May 9, 2023

The Academic Editor, PLOS ONE,

Dear Dr. Mabel Kamweli Aworh, 

RE: Point-by-point response to reviewers – Source-tracking ESBL-producing bacteria at the maternity ward of Mulago hospital, Uganda (PONE-D-23-06934) 

We thank you for considering our manuscript, and for forwarding to peer-reviewers for critique. We are indebted for the comments/suggestions and have carefully considered all of them – including comments in the attachments on email, and improved the manuscript accordingly.

Kindly find our point-by-point response to each concern, below. We outline how the comments have been addressed and point to specific line numbers and pages where to find the changes.

JOURNAL REQUIREMENTS

Response: 

The revised manuscript meets PLOS ONE’s style requirements. 

2. Thank you for stating the following in the Acknowledgments Section of your manuscript: "Special thanks to staff at the Departments of Immunology and Molecular Biology and Medical Microbiology, Makerere University College of Health Sciences for the support they provided during the time the research was conducted, as well as the research participants who agreed to be part of this study. Finally, special thanks to the Cambridge group that collected the isolates and made them available to us. AM is a Chancellor’s Fellow at the Roslin Institute and his time was paid core funding and as a Future leader fellow funded by BBSRC (BB/P007767/1) and Wellcome Trust ISSF3 (IS3-R1.09 19/20).”

Please remove any funding-related text from the manuscript and let us know how you would like to update your Funding Statement. Currently, your Funding Statement reads as follows: "The author(s) received no specific funding for this work."

Response: 

We have removed funding-related text from the revised manuscript. 

Kindly update our Funding Statement as; 

"The author(s) received no specific funding for this work. AM is a Chancellor’s Fellow at the Roslin Institute and his time was paid core funding and as a Future leader fellow funded by BBSRC (BB/P007767/1) and Wellcome Trust ISSF3 (IS3-R1.09 19/20)." 

As advised, we have included the above statement in the cover letter for you to change the online submission on our behalf. 

Response: 

All the cited references have been cross checked online and to the best of our knowledge, none has been retracted. We realize that some journals might be no longer in production, for example SpringerPlus that published article No. 23 was discontinued however, the article itself is not retracted – it appears online and is searchable. 

RESPONSE TO REVIEWERS' COMMENTS:

REVIEWER #1

This is a very informative and technical work. Few inputs made; work is good for publishing. I appreciate the fact that you numbered your lines making reviewing this work a lot easier, though some numbered lines are blank I would use your numbering as is.

Response: We are delighted to know that you found our manuscript informative and worth publishing in PLOS ONE. We have dully addressed all the concerns you raised; please find our response below;

Comment 1: Line 34, page 9; please correct cross-section study to cross-sectional study

Response: Thank you – this has been corrected, see line 34. 

Comment 2: Lines 50 - 52, page 10; please review this statement and clarify.

Response: Apologies for our not being clear here. All we intended to say is that we employed a data analysis procedure referred to as ‘spatial cluster analysis’, in which we investigated the phenotypic and genotypic drug susceptibility characteristics of the isolates and inferred relationships among them. Accordingly, the statement has been rephrased for clarity – it now reads as “Additionally, cluster analysis revealed that isolates from mothers, newborns, health workers, and environment have similar phenotypic and genotypic characteristics, suggesting transmission of multidrug-resistant EKE bacteria to newborns” (lines 51-53). For a better perspective on this, we also reviewed the methods section to read ‘To infer relationships among the EKE isolates, spatial cluster analysis of the phenotypic and genotypic antibiotic susceptibility profiles was done using the Ridom server.’, see lines 40-41. 

Comment 3: Lines 114 – 115, page 12; do you have access to more recent birth records? If yes, please use that unless you have a good reason for using these from 9 years ago.

Response: Apologies – we are unable to obtain the most up-to-date birth records for Mulago hospital at this moment. The information we shared is based on data made available to the public at that point in time.

Comment 4: Line 132, page 13; please do not abbreviate the word approximately

Response: The word approximately is now written in full in the revised manuscript.

Comment 5: Line 137, page 14; please change carried out from to “carried out in “

Response: As suggested, the phrase ‘Carried out from’ has been changed to ‘carried out in’ (line 143)

Comment 6: Line 240, page 18; please use chi 2 or chi square

Response: chi2 has been changed to chi square, line 245 

Comment 7: Lines 242 - 243, page 18; please change was to were

Response: Here we are referring to ‘cluster analysis’ hence we have maintained ‘was’ as it is singular; however, we have rephrased for clarity, see lines 246-248.

Comment 8: Include your keywords to the written manuscript

Response: Keywords have been included, line 60

Comment 9: The omission of gloves used by health workers as part of where samples should be taken was a huge oversight in this work nevertheless much work was put into this study and it is very technical and highly informative. The work itself has a lot of papers embedded

Response: We concur with the reviewer's assessment that the failure to include information about the gloves used by healthcare workers in relation to where samples were taken was a significant oversight in our research. As we drew our data from a parent study that did not account for this information, it was not available for our analysis. We appreciated your understanding of this limitation, which we indeed acknowledged and will consider in further studies.

Comment 10: Justice was done to the title

Response: We thank you for your appreciation.

REVIEWER #2 

Comment (Abstract / Introduction): 

• Thank you for this important study that characterizes the sources of ESBL which is crucial for infection control and prevention at a tertiary hospital ward in Uganda.

• The Abstract is well written, however, the authors should consider adding the policy or practice implication to the abstract. Additionally, line 24 reads cross-section instead of cross-sectional.

• Major strengths of this study include the well-written methodology and the presentation of the result findings in tables that are clear and easy to understand. A major weakness of this study is the time that the data was collected which was more than five years ago. however, the study still provides important insights.

• The Introduction gives a good background and a good justification for the study.

Response: 

We thank you for your comments and for your appreciation of the important insights our manuscript provides. As you have advised, the policy / practice implication has been provided to the revised abstract, see lines 56 to 59. Also, in agreement with REVIEWER 1, we have changed ‘cross-section’ to ‘cross-sectional’, line 34.

Comment (Methods):

• Line 108: Consider removing "isolates" as there is no description of isolates in this section.

• Line 125: The authors should consider consistency in describing the sample collection sites e.g. nasal is referred to nares and anterior nares in different sections; armpit is referred to as axillar (line 264; also note that the correct word is axilla)

• Line 127: The authors should consider changing "as well" to other phrases such as Also, Additionally etc.

• Line 132: approx. should be be written in full or changed to another term. Same as for Line 161.

• Line 173: Consider adding a 'comma' after "ruler"

Response:

• The word “isolates” has been removed as suggested, line 113 

• In this revision, “nare” has been replaced with “nose” and “axilla” has been replaced with “armpit”, see Table 1. Also, this reminded us of the informed consent procedure and documents, in which study participants (mothers) were told that a swab will be picked from their nose and armpit.

• The phrase “as well” has been replaced with “additionally”, line 132

• In the revised version, the word “approx.” has been written in full throughout the text

• A coma has been added after ‘ruler’, see line 179.

Comment (Results and Discussion):

• Line 251: The authors should consider writing "discussion" as a sentence case

• Line 261: further should be in the past tense.

• Line 328: The authors should consider changing "as well" to other phrases such as Also, Additionally etc.

• Line 360: Kayange et al should be written as Kayange et al., and the year of the article should be included.

Response:

• As suggested, discussion has been written as sentence case, line 256. 

• As suggested, the sentence now reads as “Other Gram-negative bacteria were identified but not discussed further”, line 265-267.

• The “As well” phrase has been replaced with “Additionally”, see line 334.

• The suggested change has been made, see line 369.

Comment (Limitations):

• Line 387: The authors should consider writing the none the less as one word.

• Also consider acknowledging that the data was collected in 2015-2016 and any implication if any.

Response:

• The phrase “none the less” has been rewritten as one word i.e., “nonetheless”, line 396. 

• We thank you; the acknowledgement of data collection in 2015-2016 and implication thereof, have been included, see lines 397-402

Comment (Conclusions):

• Line 396: Consider reworking "carrying out more studies" to "more studies should be conducted". Also consider rephrasing "urgent infection control protocol" as this data for this study was collected between 2015-2016, and a lot of things might have changed.

• Overall, this was a well conducted and reported study with necessary ethical approvals obtained.

Response:

• The phrase "carrying out more studies" has been changed to "more studies should be conducted", line 410-411. 

• Further, in light of your suggestion the phrase "urgent infection control protocol" has been revised and toned down, see lines 405-414.

REVIEWER #3

Comment: 

• The manuscript was well laid out with a good method section; However, authors need to explain how they arrived at a sample of 137 women? is it a purposive sampling technique? 

• Religion of participants were listed in Table 1, and it is unclear how that relates to having ESBL producing bacteria. Consider dropping religion.

Response: 

• Thank you for mentioning that our manuscript is well-laid out. As you have hinted, for sample size we used purposive sampling to select the 137 pregnant women scheduled for caesarian surgical delivery. So, in the revised manuscript we have added a statement to this effect, see lines 129 & 258.

• As advised, data on religion has been removed, see Table 1.

---

## [Decision Letter · Decision Letter 1]

26 May 2023

Source-tracking ESBL-producing bacteria at the maternity ward of Mulago hospital, Uganda

PONE-D-23-06934R1

Dear Dr. KATEETE,

We’re pleased to inform you that your manuscript has been judged scientifically suitable for publication and will be formally accepted for publication once it meets all outstanding technical requirements.

Kind regards,

Mabel Kamweli Aworh, DVM, MPH, PhD. FCVSN

Academic Editor

PLOS ONE

Additional Editor Comments (optional):

Reviewers' comments:

Reviewer's Responses to Questions

**Comments to the Author**

1. If the authors have adequately addressed your comments raised in a previous round of review and you feel that this manuscript is now acceptable for publication, you may indicate that here to bypass the “Comments to the Author” section, enter your conflict of interest statement in the “Confidential to Editor” section, and submit your "Accept" recommendation.

Reviewer #1: All comments have been addressed

Reviewer #2: All comments have been addressed

Reviewer #3: All comments have been addressed

2. Is the manuscript technically sound, and do the data support the conclusions?

Reviewer #1: Yes

Reviewer #2: (No Response)

Reviewer #3: Yes

3. Has the statistical analysis been performed appropriately and rigorously? 

Reviewer #1: Yes

Reviewer #2: (No Response)

Reviewer #3: Yes

4. Have the authors made all data underlying the findings in their manuscript fully available?

Reviewer #1: Yes

Reviewer #2: (No Response)

Reviewer #3: Yes

5. Is the manuscript presented in an intelligible fashion and written in standard English?

Reviewer #1: Yes

Reviewer #2: (No Response)

Reviewer #3: Yes

6. Review Comments to the Author

Reviewer #1: All previous recommendations by reviewers have been corrected accordingly hence good for publishing. It is a good work with good information for health facilities, health care workers and IPC enthusiasts . The experiments were made clearly with ability for replication.

Reviewer #2: (No Response)

Reviewer #3: (No Response)

7. PLOS authors have the option to publish the peer review history of their article (what does this mean?). If published, this will include your full peer review and any attached files.

Reviewer #1: **Yes: **Folashade Onatola Bamidele

Reviewer #2: No

Reviewer #3: **Yes: **Augustine Olajide Dada

---

## [Editor Report · Acceptance letter]

1 Jun 2023

PONE-D-23-06934R1 

Source-tracking ESBL-producing bacteria at the maternity ward of Mulago hospital, Uganda 

Dear Dr. Kateete:

I'm pleased to inform you that your manuscript has been deemed suitable for publication in PLOS ONE. Congratulations! Your manuscript is now with our production department. 

Kind regards, 

on behalf of

Dr. Mabel Kamweli Aworh 

Academic Editor

PLOS ONE